# REFLECTIVE FLOW SAMPLING ENHANCEMENT

## ABSTRACT

The growing demand for text-to-image generation has led to rapid advances in generative modeling. Recently, text-to-image diffusion models trained with flow matching algorithms, such as FLUX, have achieved remarkable progress and emerged as strong alternatives to conventional diffusion models. At the same time, inference-time enhancement strategies have been shown to improve the generation quality and text–prompt alignment of text-to-image diffusion models. However, these techniques are mainly applicable to conventional diffusion models and usually fail to perform well on flow models. To bridge this gap, we propose Reflective Flow Sampling (RF-Sampling), a novel training-free inference enhancement framework explicitly designed for flow models, especially for the CFG-distilled variants (i.e., models distilled from CFG guidance techniques) like FLUX. RF-Sampling leverages a linear combination of textual representations and integrates them with flow inversion, allowing the model to explore noise spaces that are more consistent with the input prompt. This approach provides a flexible and effective means of enhancing inference without relying on CFG-specific mechanisms. Extensive experiments across multiple benchmarks demonstrate that RF-Sampling consistently improves both generation quality and prompt alignment, whereas existing state-of-the-art inference enhancement methods such as Z-Sampling fail to apply. Moreover, RF-Sampling is also the first inference enhancement method that can exhibit test-time scaling ability to some extent on FLUX.

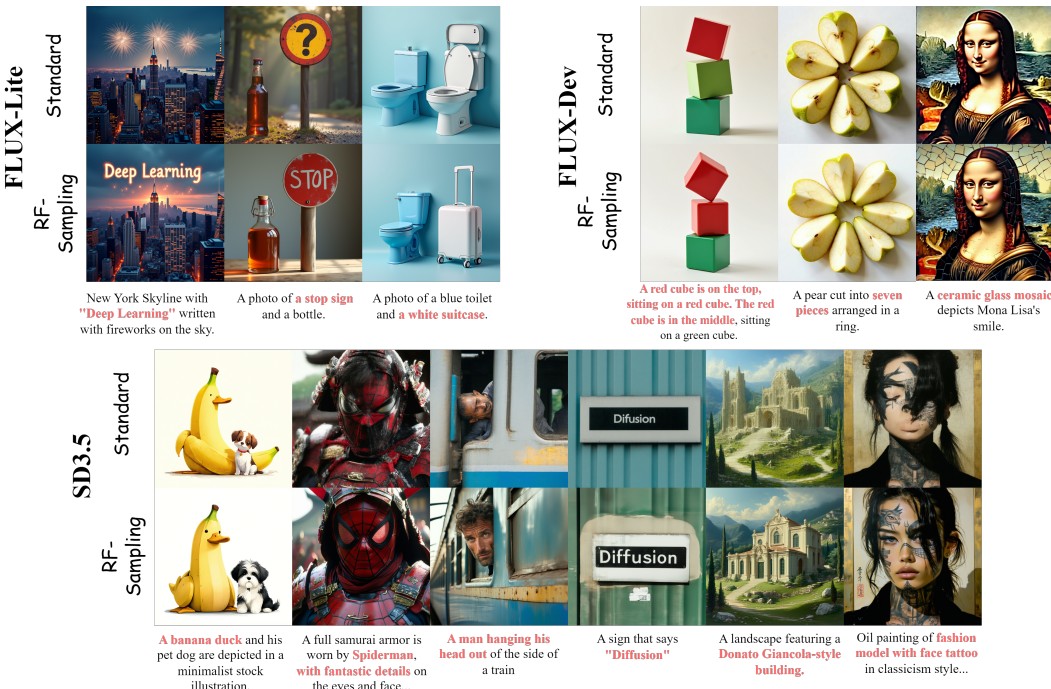

Figure 1: Qualitative comparisons with three representative flow models. Images for each prompt are synthesized using the same random seed. More visualization results are in Appendix D.

# 1 INTRODUCTION

Text-to-image (T2I) generation has become one of the most active areas in generative modeling, driven by the growing demand for creating high-quality images from natural language prompts (Rombach et al., 2022; Labs, 2024; Daniel Verdú, 2024; Esser et al., 2024). Recent advances in diffusion models and their training algorithms have led to remarkable progress, enabling strong performance across diverse domains (Yang et al., 2023; Esser et al., 2024; Lipman et al., 2022; Liu et al., 2022b; Ho et al., 2020). To further improve generation quality and prompt alignment, a variety of inference enhancement methods have been proposed for diffusion models (Singhal et al., 2025; Ma et al., 2025b; Ho & Salimans, 2022). Among them, inversion-based techniques such as Z-Sampling (Bai et al., 2025a) exploit the discrepancy of the Classifier-Free Guidance (CFG) (Ho & Salimans, 2022) parameter between denoising and DDIM inversion (Song et al., 2023a), while recent weak-to-strong methods like W2SD-Sampling (Bai et al., 2025b) amplify semantic information hidden in the noise latent to achieve state-of-the-art performance. These strategies demonstrate the effectiveness of inference-time interventions for diffusion denoising processes.

At the same time, T2I diffusion models trained with flow matching algorithms (Lipman et al., 2022), such as FLUX (Labs, 2024; Daniel Verdú, 2024), have recently emerged as promising alternatives to conventional diffusion models, offering both competitive quality and efficient sampling. However, most existing inference enhancement methods are tightly coupled with conventional diffusion-specific mechanism and fail to generalize to flow models. To mitigate this limitation, recent work such as CFG-Zero* (Fan et al., 2025) has proposed optimized scaling and zero-init strategies to adapt CFG-style guidance to flow matching. Nevertheless, the reliance on CFG-specific techniques still restricts the broader applicability of inference enhancement strategies, especially as CFG-distilled variants (Meng et al., 2023), such as FLUX, continue to gain traction as efficient T2I generators.

To fill this gap, we introduce **Reflective Flow Sampling (RF-Sampling)**, a novel training-free inference enhance framework explicitly designed for flow models that bypasses the reliance on CFG-style guidance entirely. Inspired by the key findings that rich semantic noise latent can improve the generative ability of conventional diffusion model (Wang et al., 2024; Bai et al., 2025a; Zhou et al., 2025; Po-Yuan et al., 2023), our key idea is to interpolate textual representations and integrate them with flow inversion, which allows the model to explore noise spaces that are more consistent with the input prompt. We refer to such flow inversion as reflective flow, motivated by the term "diffusion reflection" (Bai et al., 2025a). Our reflective flow mechanism provides a flexible, scalable, and effective way to enhance inference without relying on CFG-specific mechanisms, making it widely applicable across flow models, especially CFG-distilled variants, like FLUX.

This paper validates the significant effectiveness of RF-Sampling through extensive experiments on multiple benchmarks. Our method consistently enhances both image quality and text–prompt alignment across different settings, achieving top-1 performance in evaluations conducted by diverse human preference models (Schuhmann; Xu et al., 2023; Kirstain et al., 2023; Wu et al., 2023). To provide an intuitive illustration of this improvement, a representative visualization is shown in Fig. 1. The images synthesized by RF-Sampling demonstrate a noticeable improvement in overall quality, aesthetic style, and semantic faithfulness, along with numerical improvements. In contrast, our experiments show that existing inference enhancement methods do not perform well and the state-of-the-art diffusion-based inference enhancement methods cannot be directly applied to flow models, highlighting the necessity of our approach. As Fig. 2 suggests, RF-Sampling is also the first inference enhancement method that can exhibit test-time scaling ability to some extent on FLUX. Moreover, we extend our method to several tasks like lora combination, image editing and video synthesis to demonstrate the scalability of RF-Sampling.

# 2 RELATED WORK

## 2.1 TEXT-TO-IMAGE GENERATION

T2I generation is a rapidly evolving branch of generative modeling, aiming to synthesize realistic images that align with given textual descriptions. Early methods primarily relied on autoregressive models (Salimans et al., 2017; Chen et al., 2020) or generative adversarial networks (Goodfellow et al., 2014; Mirza & Osindero, 2014). However, in recent years, Diffusion models (Ho et al., 2020; Rombach et al., 2022) have emerged as the dominant paradigm in T2I due to their ability to generate

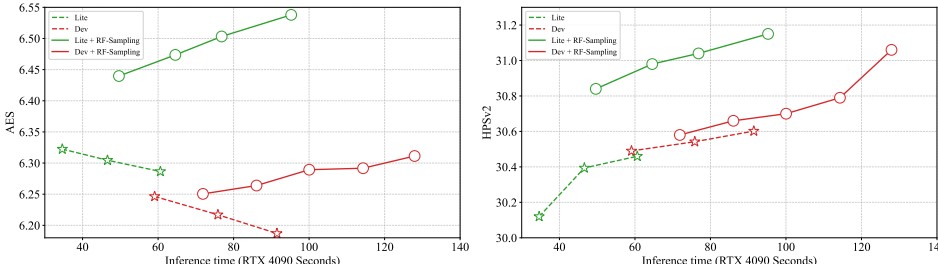

Figure 2: RF-Sampling outperforms standard sampling with the same time consumption and significantly enhances the performance of FLUX-Lite and FLUX-Dev. With the increase of inference time, RF-Sampling consistently performs well, validating the scalability of our method.

high-quality and high-resolution images. These models generate images through a stepwise denoising process, starting from a random noise image and gradually transforming it into a meaningful image. In addition to conventional diffusion models, Flow Matching (Lipman et al., 2022; Liu et al., 2022b) is an emerging diffusion model training technique that has rapidly gained traction as a strong alternative. Flow matching learns a continuous transformation that smoothly maps a simple noise distribution to the data distribution via matching the velocity. Unlike conventional diffusion models, which require multiple discrete denoising steps, flow matching models such as FLUX (Labs, 2024; Daniel Verdú, 2024) can achieve efficient sampling with fewer neural function evaluations (NFEs), significantly reducing inference time while maintaining comparable, even superior generation quality to top conventional diffusion models. This efficiency advantage makes flow matching models particularly attractive for applications requiring fast generation. Our work focuses on developing dedicated inference enhancement strategies for these efficient flow models.

## 2.2 CFG-DISTILLED GUIDANCE

Classifier-Free Guidance (Ho & Salimans, 2021) has become a foundational technique in conditional diffusion models, as it improves alignment between synthesized images and text prompts by blending conditional and unconditional outputs during inference. Despite its effectiveness, CFG doubles inference cost by requiring two forward passes per denoising step.

To mitigate this inefficiency, a class of methods termed *CFG-distilled* (Meng et al., 2023; Li et al.) techniques has been proposed. These methods aim to replicate the benefits of CFG using a single forward pass, thereby maintaining alignment quality while significantly reducing computational overhead. From a deployment standpoint, CFG-distilled models, such as FLUX (Daniel Verdú, 2024; Labs et al., 2025) are particularly crucial: they preserve the alignment advantages of CFG while drastically improving inference speed, making real-time or on-device applications feasible without prohibitive computation requirements.

## 2.3 INFERENCE ENHANCEMENT FOR T2I GENERATION

To enhance the generation quality and text alignment of conventional diffusion models, researchers have explored a range of inference enhancement strategies, which can be applied to pretrained models without requiring additional training. One key enhancement technique is Z-Sampling (Bai et al., 2025a), which leverages differences in the CFG parameters during the denoising process and DDIM inversion (Song et al., 2023a) to enhance the generation, suggesting that the noise latent space holds rich semantic information crucial for image quality. Similarly, W2SD-Sampling (Bai et al., 2025b) utilizes weak-to-strong techniques to enhance semantic information in the noise latent space, achieving state-of-the-art performance. Other methods, such as (Singhal et al., 2025; Ma et al., 2025b; Wang et al., 2024; Zhou et al., 2025; Po-Yuan et al., 2023), have also explored improving generation by manipulating the noise or latent space, indicating that intervention at the inference stage is an effective direction. Furthermore, in the context of Flow Matching, CFG-Zero* (Fan et al., 2025) mitigates the shortcomings of Flow CFG (Zheng et al., 2023) by incorporating an optimized scale and zero-init, thereby refining the inference trajectory. Despite the significant success of these inference enhancement strategies, they are typically tailored to the conventional diffusion models or rely on specific inference mechanisms, such as CFG technique and particular inversion algorithms.

As a result, these methods cannot be directly transferred to flow models, especially when dealing with CFG-distilled variants. This limitation is particularly pressing as flow models gain increasing popularity due to their efficiency advantages, making it crucial to address this gap.

## 3 METHOD

In this section, we discuss how to encode semantic information into latents through the prompt embedding gap and derive the formulation of RF-Sampling.

### 3.1 FLOW MATCHING MODELS

Flow matching models represent a new class of generative models that synthesize images by solving an ordinary differential equation(ODE). The core idea is to train a neural network, parameterized as a vector field $v_\theta(x, t)$, to predict the flow that pushes a simple prior distribution $p_0(x)$ (e.g., standard Gaussian) to a complex target data distribution $p_1(x)$. The inference process then involves sampling a point from the prior $x_0 \sim p_0(x)$ and solving the ODE:

$$\frac{dx}{dt} = v_\theta(x, t), \tag{1}$$

from $t = 0$ to $t = 1$ to obtain the final generated sample $x_1$. For convenience, we refer to this class of models as *flow models* throughout the paper.

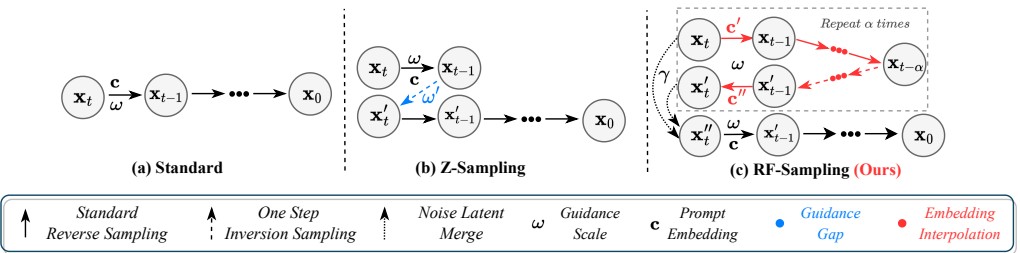

Figure 3: Illustration of RF-Sampling. Compared to previous methods, RF-Sampling employs interpolation on text embeddings similar to the traditional CFG, thereby enhancing the model's generation quality and making it more suitable for flow diffusion models, especially CFG-distilled models.

### 3.2 TEXT EMBEDDING FOR DIFFERENT PROCESS

For T2I generation, the vector field is conditioned on a text embedding $c$, denoted as $v_\theta(x, t, c)$. Unlike conventional diffusion models, where CFG relies on joint training with both conditional and unconditional branches (Ho & Salimans, 2021; Fan et al., 2025), Some flow models are typically trained only under conditional settings (Labs, 2024; Daniel Verdú, 2024). As a result, directly using CFG techniques or adopting an empty-text embedding as guidance for this kind of CFG-distilled flow models is inappropriate. To address this, we employ a linear interpolation between the conditional text embedding $c_{text}$ and an unconditional empty-text embedding $c_{uncond}$, yielding a mixed text embedding $c_{mix}$. In addition, we introduce a the amplifying weight $s$ to explicitly amplify the semantic discrepancy arising from the different text embeddings used in the denoising and inversion processes. The combination of text embedding can be described as:

$$c_{mix} = \beta \cdot c_{text} + (1 - \beta) \cdot c_{uncond},$$
$$c_w = c_{text} + s \cdot c_{mix}, \tag{2}$$

where $\beta$ is the interpolation weight directly controlling the difference between text prompt embeddings. A higher $\beta$ typically leads to a stronger alignment with the prompt. Therefore, the combination of $\beta$ and $s$ enables us to adjust the degree of text guidance throughout the inference process.

## 3.3 Reflective Flow Sampling

Building on the findings of Z-Sampling (Bai et al., 2025a), which demonstrated that diffusion models can accumulate prompt-related semantics in the latent noise by applying strong CFG guidance during denoising and weaker CFG guidance during inversion, we propose a novel approach that bypasses CFG guidance using a linear interpolation of textual representations. Specifically, our method enables flow models to adopt stronger semantic guidance in the denoising phase while applying weaker semantic influence during inversion. This design facilitates the accumulation of semantic information in the latent noise, thereby improving both image quality and text–image alignment. Our approach introduces a three-stage "reflection" loop within each integration step of the ODE solver as shown in Fig. 3. The core idea is to leverage a low-weight semantic guidance inversion to correct the trajectory, ensuring the generated latent features remain in a semantically rich region.

Let $x_t$ be the latent feature at time step $t$, the standard text embedding $c$, and the standard guidance scale $w$. Our method proceeds as follows for each step:

**Stage 1: High-Weight Denoising**  First, we perform a standard denoising step using a relatively **high interpolation weight** $\beta_{high}$ and a relatively **high amplifying weight** $s_{high}$ to get the mixed text embedding $c'$, according to Eqn. 2. We then take $\alpha$ steps of the ODE solver from $t$ to $t - \alpha$ to obtain the next latent feature $x_{t-\alpha}$:

$$x_{t-\alpha} = x_t + \sum_{i=1}^{\alpha} v_\theta(x_{t-i+1}, t-i+1, c')\Delta t \tag{3}$$

where $v_\theta$ is the conditioned vector field, $\alpha$ is the forward steps, and $\Delta t$ is the integration step size. This stage ensures a rapid and strong alignment with the given text prompt.

**Stage 2: Low-Weight Inversion**  This is the key stage of our method. Instead of directly using the newly obtained $x_{t-\alpha}$, we perform a backward-step ODE solving from $x_{t-\alpha}$. Crucially, this inversion uses a **low interpolation weight** $\beta_{low}$ and a relatively **low amplifying weight** $s_{low}$ for the mixed text embedding $c''$, according to Eqn. 2. The corrected latent feature $x'_t$ is obtained by:

$$x'_t = x_{t-\alpha} - \sum_{i=1}^{\alpha} v_\theta(x_{t-\alpha+i-1}, t-\alpha+i-1, c'')\Delta t \tag{4}$$

where $x'_t$ is the corrected latent feature after inversion. This backward step effectively "reflects" the high-weight-guided latent feature back towards a more semantically centered region of the latent space. It filters out potential latent that have rich semantic information, providing a more text information starting point for the next forward step.

**Stage 3: Normal-Weight Denoising**  With the semantically corrected feature $x'_t$, we proceed with the final denoising step for this time interval. In order to stabilize the denoising process, we balance the weights between $x_t$ and $x'_t$ using **merge ratio** $\gamma$. Then we utilize the standard text embedding $c$ and the standard guidance scale $w$ to obtain the final latent feature for the next time step $x'_{t-1}$:

$$\begin{aligned} x''_t &= x_t + \gamma \cdot (x_t - x'_t), \\ x''_{t-1} &= x''_t + v_\theta(x''_t, t, c)\Delta t \end{aligned} \tag{5}$$

where $x''_{t-1}$ is the final latent feature for the next time step. This step ensures that the generation process continues to progress towards the target image distribution with an appropriate level of text alignment, building on the refined latent feature from the inversion stage.

By repeating this three-stage process for each time step, RF-Sampling achieves a better high-quality and semantically coherent image synthesis. The detail process is shown in Algorithm 1.

## 4 Experiment

### 4.1 Experiment Setting

We conduct a comprehensive evaluation of several T2I diffusion models. For a detailed description of the benchmarks, evaluation metrics, model architectures and hyperparameter settings, please refer to Appendix B. Below is a summary of our experimental setup.

**Benchmarks.** Our evaluation leverages several established benchmarks to assess a wide range of capabilities. For human preference alignment, we use **Pick-a-Pic** (Kirstain et al., 2023) and **HPD v2** (Wu et al., 2023). To evaluate compositional reasoning, we employ **DrawBench** (Saharia et al., 2022), **GenEval** (Ghosh et al., 2023), and **T2I-Compbench** (Huang et al., 2023). For text-to-video (T2V) and in-context image generation, we utilize **ChronoMagic-Bench-150** (Yuan et al., 2024) and **FLUX-Kontext-Bench** (Labs et al., 2025), respectively.

**Evaluation Metrics.** To quantify model performance, we utilize several metrics designed to reflect human perception. These include **PickScore** (Kirstain et al., 2023), **HPS v2** (Wu et al., 2023), and **ImageReward** (Xu et al., 2023) for measuring alignment with human preferences, and the **Aesthetic Score (AES)** (Schuhmann) for assessing visual appeal. For T2V evaluation on ChronoMagic-Bench-150, we use UMT-FVD, UMTScore, GPT4o-MTScore, and MTScore.

**Flow Models.** Our analysis focuses on five state-of-the-art flow models. For T2I generation, we evaluate **FLUX-Dev** (Labs, 2024), its lightweight variant **FLUX-Lite** (Daniel Verdú, 2024), and **StableDiffusion-3.5** (Esser et al., 2024). For T2V generation, we use **Wan2.1-T2V-1.3B** (Wan et al., 2025), and for in-context image editing, we evaluate **FLUX-Kontext** (Labs et al., 2025).

## 4.2 MAIN EXPERIMENT

Table 1: Main experiments on HPDv2 (Wu et al., 2023) dataset across 3 different models. The experiments show the consistent superior performance compared with previous methods, highlighting the effectiveness of our RF-Sampling. Note that other baselines are not applicable to FLUX.

| Model | Method | Animation | | Concept-art | | Painting | | Photo | | Average | |
|---|---|---|---|---|---|---|---|---|---|---|---|
| | | AES(↑) | HPSv2(↑) | AES(↑) | HPSv2(↑) | AES(↑) | HPSv2(↑) | AES(↑) | HPSv2(↑) | AES(↑) | HPSv2(↑) |
| | Standard | 5.9474 | 30.93 | 6.1926 | 28.59 | 6.4161 | 28.84 | 5.4077 | 27.66 | 5.9909 | 29.01 |
| | GI (Kynkäänniemi et al., 2024) | 5.9814 | 26.23 | 6.2188 | 23.48 | 6.2188 | 23.61 | 5.3417 | 23.81 | 5.9401 | 24.28 |
| SD3.5 | Z-Sampling (Bai et al., 2025a) | 5.8729 | 30.58 | 6.0427 | 27.58 | 6.2579 | 28.21 | 5.4394 | 27.92 | 5.9032 | 28.57 |
| (28 steps) | CFG++ (Chung et al., 2024) | 5.8329 | 29.81 | 6.0969 | 27.41 | 6.3206 | 27.81 | 5.3969 | 27.04 | 5.9118 | 28.02 |
| | CFG-Zero* (Fan et al., 2025) | 5.9743 | 31.22 | 6.2066 | 29.27 | 6.4280 | 29.22 | 5.4190 | 27.65 | 6.0061 | 29.34 |
| | RF-Sampling | 6.0164 | 31.71 | 6.2093 | 29.80 | 6.3702 | 29.77 | 5.4973 | 28.51 | 6.0243 | 29.80 |
| FLUX-Lite | Standard | 6.2635 | 31.96 | 6.5378 | 30.01 | 6.7381 | 30.67 | 5.8132 | 29.04 | 6.3381 | 30.42 |
| (28 steps) | RF-Sampling | 6.4350 | 32.78 | 6.6240 | 30.70 | 6.7832 | 30.95 | 5.9864 | 29.93 | 6.4572 | 31.09 |
| FLUX-Dev | Standard | 6.1459 | 32.26 | 6.4934 | 30.56 | 6.4934 | 31.27 | 5.6515 | 29.64 | 6.1960 | 30.93 |
| (50 steps) | RF-Sampling | 6.1866 | 32.40 | 6.5153 | 30.80 | 6.5153 | 31.45 | 5.6799 | 29.81 | 6.2243 | 31.12 |

Table 2: Main experiments on Pick-a-Pic (Kirstain et al., 2023) and DrawBench (Saharia et al., 2022) datasets across 3 different models. Obviously, our proposed RF-Sampling exhibits superior performance across 4 different metrics. Note that other baselines are not applicable to FLUX.

| Model | Method | Pick-a-Pic | | | | DrawBench | | | |
|---|---|---|---|---|---|---|---|---|---|
| | | PickScore(↑) | ImageReward(↑) | AES(↑) | HPSv2(↑) | PickScore(↑) | ImageReward(↑) | AES(↑) | HPSv2(↑) |
| | Standard | 21.99 | 85.13 | 5.9435 | 29.32 | 22.60 | 86.02 | 5.4591 | 27.76 |
| | GI | 21.19 | 28.94 | 5.9534 | 24.63 | 22.11 | 47.53 | 5.4279 | 23.96 |
| SD3.5 | Z-Sampling | 21.73 | 89.03 | 5.9091 | 28.84 | 22.55 | 92.05 | 5.4784 | 28.06 |
| (28 steps) | CFG++ | 21.79 | 85.17 | 5.8821 | 28.50 | 22.54 | 81.80 | 5.3757 | 27.18 |
| | CFG-Zero* | 21.88 | 86.78 | 5.9536 | 29.37 | 22.66 | 91.90 | 5.4511 | 28.10 |
| | RF-Sampling | 21.99 | 101.50 | 5.9981 | 29.90 | 22.64 | 94.10 | 5.4915 | 28.74 |
| FLUX-Lite | Standard | 21.91 | 86.64 | 6.3224 | 30.12 | 22.59 | 86.51 | 6.2635 | 31.96 |
| (28 steps) | RF-Sampling | 22.05 | 99.21 | 6.5379 | 31.16 | 22.69 | 96.15 | 6.4350 | 32.79 |
| FLUX-Dev | Standard | 22.06 | 97.47 | 6.2464 | 30.49 | 22.84 | 99.73 | 6.1459 | 32.39 |
| (50 steps) | RF-Sampling | 22.19 | 100.90 | 6.3113 | 31.06 | 22.93 | 106.21 | 6.1866 | 32.40 |

To validate the effectiveness of our method, we conduct evaluations using multiple human preference models which score the images generated by our approach. Since prior inference enhancement methods rely on CFG technique, they cannot be applied to CFG-distilled flow models. To further validate our idea, we conduct additional analyses in the appendix. The results, as illustrated in Fig. 6, Fig. 11 and Fig. 12, reveal that previous methods tend to cause the generated images to deviate from the true data distribution, while RF-Sampling trajectories consistently demonstrate strong convergence towards the real data distribution. Therefore, we use standard sampling as the baseline for FLUX. The results in Tab. 1 and Tab. 2 prove that our method consistently achieves top-1 performance across most metrics. In addition, we report preference-winning rate experiments among different human preference models in Fig. 4 and Fig. 5, where our method achieves up to

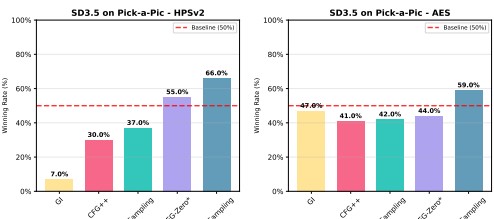 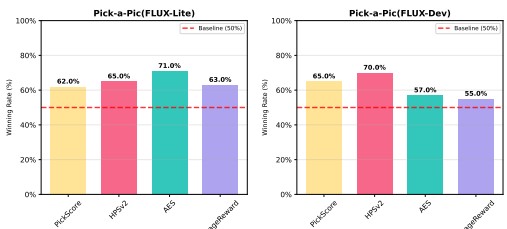

Figure 4: The winning rate of RF-Sampling over other methods on SD3.5. The standard sampling (baseline) winning rate defaults to 50%. The results reveal the superiority of RF-Sampling in synthesizing images with good quality.

Figure 5: The winning rate of RF-Sampling over other methods on FLUX. The standard sampling (baseline) winning rate defaults to 50%. The results reveal the superiority of RF-Sampling in synthesizing images with good quality.

70% winning rate under certain expert preferences. Moreover, we evaluate our method on the T2I and GenEval benchmarks to demonstrate its effectiveness. The corresponding results are provided in the appendix, as shown in Tab. 6 and Tab. 7. To highlight the advantages of our approach, we provide qualitative visualizations in Fig. 1, with additional synthesized examples in Appendix Sec. D. These visualizations further highlight the enhanced inference capability of our method.

## 4.3 Ablation Studies and Additional Analysis.

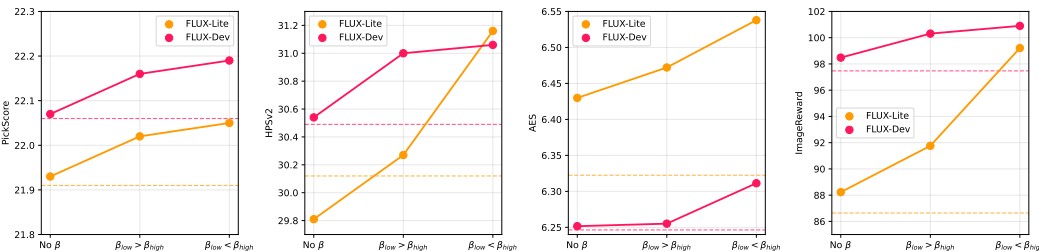

Figure 7: Ablation study on the effect of $\beta_{low}$ and $\beta_{high}$. No $\beta$ means that we do not implement the interpolation weight in Eqn. 2. The results reveal that following the high-weight denoising → low-weight inversion paradigm can enchance the quality of synthesized images. The dotted lines represents the performance of the standard method. This indicates that within a certain range of values, RF-Sampling perform better than the standard one, demonstrating the robustness of it.

To better highlight the characteristics of our method, we conducted extensive quantitative and qualitative experiments, as presented below. More results are provided in Appendix C.

**High denoising and low inversion.** To validate the rationale behind the choice of the interpolation parameter $\beta$, we conduct experiments with different settings of $\beta$. The results, shown in Fig. 7, confirm the effectiveness of interpolation and justify assigning higher weights to the forward process while using lower weights for the inverse process. As a complement, to provide a more intuitive understanding of the effect of varying $\beta$, we present the corresponding visualizations in Fig. 13 and Fig. 14. In addition, to examine the effectiveness of parameter $s$ in amplifying the semantic gap, we perform experiments as illustrated in Fig. 8. The results

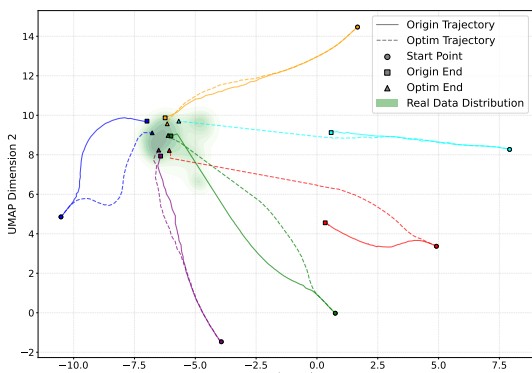

Figure 6: Visualization of the sampling trajectories sampled by our method and the standard approach. Compared with Z-Sampling and W2SD (see Fig. 11 and Fig. 12 in Appendix), RF-Sampling produces results that better align with the real data distribution.

indicate that an appropriately larger gap can better guide the model to generate high-quality images.

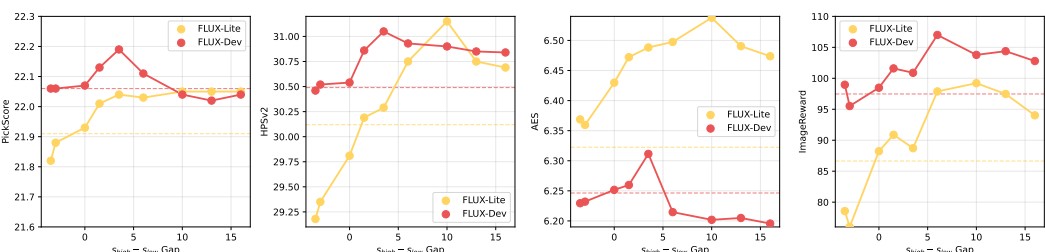

Figure 8: Ablation of the gap between $s_{high}$ and $s_{low}$. When the gap of $s_{high}$ - $s_{low}$ increases within a certain range, the quality of synthesized images improves. The dotted lines represents the performance of the standard method. This indicates that within a certain range of values, RF-Sampling perform better than the standard one, demonstrating the robustness of it.

**Optimal Steps.** To further validate the contribution of our method at each inference step, we evaluate the proportion of steps performing reflection relative to the total number of inference steps. The results, presented in Appendix Fig. 17, demonstrate that, in general, increasing the number of reflection-enhanced steps leads to higher image generation quality.

**Efficiency Analysis.** To demonstrate the efficiency of our method, we conduct performance comparison experiments under the same number of inference steps. As shown in Fig. 2, the results indicate that our method achieves better performance within the same inference steps. Furthermore, to further improve efficiency, we conduct orthogonal experiments with Nunchaku (Li* et al., 2025), a sampling acceleration method for FLUX. The results, presented in Tab. 3, show that our method can be effectively combined with such acceleration techniques, highlighting its potential for speedup.

Table 3: Orthogonal experiments with Nunchaku (Li* et al., 2025), a sampling acceleration method for FLUX. The results demonstrate the generalizability of RF-Sampling to sampling acceleration.

| Model | Method | PickScore(↑) | ImageReward(↑) | AES(↑) | HPSv2(↑) |
|---|---|---|---|---|---|
| FLUX-Lite (28 steps) | Standard | 21.91 | 86.64 | 6.3224 | 30.12 |
| | RF-Sampling | 22.05 | 99.21 | 6.5379 | 31.16 |
| | Standard + Nunchaku | 22.07 | 95.94 | 6.2303 | 30.47 |
| | RF-Sampling + Nunchaku | 22.23 | 102.35 | 6.4171 | 30.86 |
| FLUX-Dev (50 steps) | Standard | 22.06 | 97.47 | 6.2464 | 30.49 |
| | RF-Sampling | 22.19 | 100.90 | 6.3113 | 31.06 |
| | Standard + Nunchaku | 22.18 | 102.23 | 6.2203 | 30.73 |
| | RF-Sampling + Nunchaku | 22.22 | 107.46 | 6.2672 | 30.90 |

## 4.4 GENERALIZATION TO OTHER TASKS

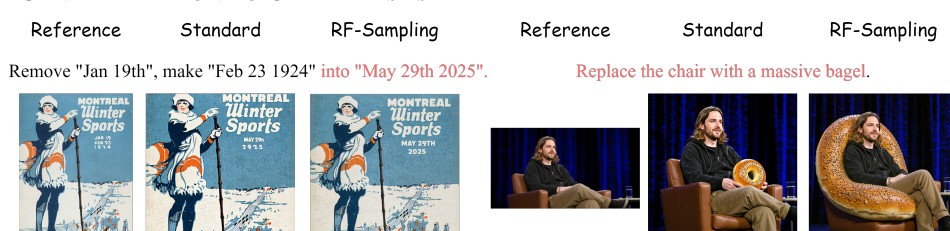

Figure 9: Image editing experiments on FLUX-Kontext Bench (Labs et al., 2025). Compared to the standard sampling, RF-sampling enables a more precise understanding of the given instruction, thereby achieving accurate image editing. For more examples, please see Appendix Fig. 29.

To further validate the generality and robustness of our approach, we extend its application beyond the standard text-to-image generation task to image editing, video generation, and LoRA fine-tuning.

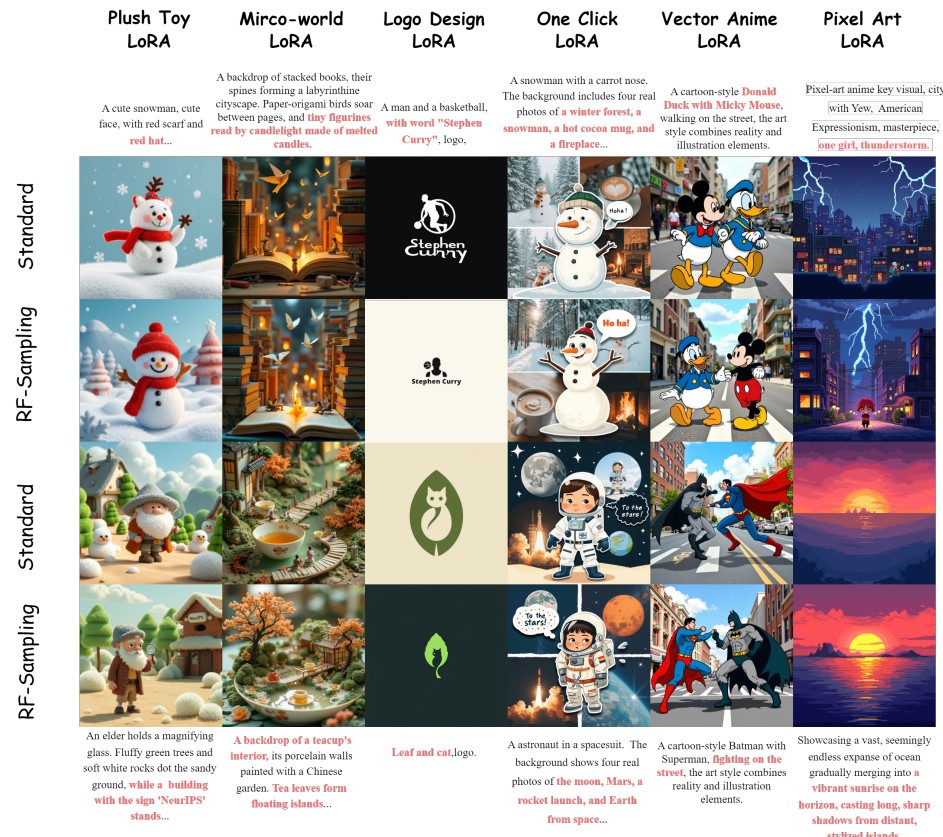

Figure 10: We combine our proposed methods with existing LoRAs in FLUX community. Our RF-Sampling can be directly applied to the corresponding downstream tasks. The synthesized images validate the effectiveness and generalizability of our method.

**Image Editing.** As shown in Fig. 9 and Appendix Fig. 29, our method achieves a winning rate of 57% when evaluated under editing scenarios, highlighting its ability to preserve semantic alignment and generate coherent modifications guided by textual instructions.

**Video Generation.** We further apply our method to the challenging task of video generation. The results, presented in Appendix Fig. 18 and Tab. 5, indicate that our approach consistently enhances video quality, confirming that the reflective mechanism generalizes well to sequential data.

**LoRA Combination.** Finally, we examine the compatibility of our method with lightweight fine-tuning techniques. As shown in Fig. 10 and Appendix Fig. 28, our method remains effective when combined with LoRA-based models, demonstrating that inference enhancements are orthogonal and complementary to parameter-efficient adaptation strategies.

## 5    CONCLUSION

In this work, we introduced RF-Sampling, a novel training-free inference enhancement method tailored for flow models, particularly those CFG-distilled variants. Our experiments demonstrate that RF-Sampling significantly improves both generation quality and text-prompt alignment, outperforming existing methods and achieving top-1 performance in various evaluations. Moreover, unlike previous inversion-based techniques, like Z-Sampling, which have been shown experimentally to cause the generated outputs to deviate from the true data distribution under guidance, RF-Sampling maintains the normal distribution while providing more consistent semantic guidance. This highlights the robustness of RF-Sampling as a reliable and flexible enhancement strategy for flow-based models. Nevertheless, the underlying mechanism behind this phenomenon remains an open question, which we leave as an important direction for future investigation.

**Ethics Statement.** We propose RF-Sampling, a training-free inference method designed to enhance the semantic faithfulness of images generated by various diffusion models, necessitates careful consideration of several ethical issues. Although RF-Sampling does not directly involve human subjects, we are committed to ensuring that its applications respect user autonomy and promote positive outcomes.

**Reproducibility statement.** We have made extensive efforts to ensure the reproducibility of our work. The full algorithmic details of RF-Sampling, including pseudo-code and parameter settings, are provided in the main paper and Appendix. All datasets used in our experiments are publicly available, and we introduce them in Sec. B in the Appendix. Finally, we will release our code in the supplementary materials.

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
