# OpenReview forum: "Reflective Flow Sampling Enhancement"
_ICLR.cc/2026/Conference — Submitted to ICLR 2026_

### Official Review · Reviewer_bGwt · 2025-10-24

**Soundness:** 2
**Presentation:** 2
**Contribution:** 2
**Rating:** 4
**Confidence:** 4

**Summary:**

This paper introduces Reflective Flow Sampling (RF-Sampling), a training-free inference-time enhancement approach for text-to-image (T2I) generation using flow-based generative models, particularly those that are CFG-distilled (such as FLUX). The method leverages an interpolation of text embeddings and a three-stage inference loop (composed of high-weight denoising, low-weight inversion, and standard denoising) to guide the generative process toward better semantic alignment with text prompts. RF-Sampling is demonstrated, through extensive experiments on multiple T2I and related tasks, to yield significant improvements in both generation quality and prompt fidelity compared to standard sampling and several baseline enhancement strategies, particularly where conventional diffusion-based techniques are inapplicable to flow models.

**Strengths:**

1. The work clearly identifies a real limitation in the applicability of inference-time enhancement techniques to flow-matching-based text-to-image models, a rising class of efficient generative models that are not well-served by prior methods.
2. The approach is mathematically formalized, with explicit equations describing the reflective sampling mechanism (see Eqs. for staged denoising/inversion, Section 3.3), and its integration with flow-based ODE solvers is well-articulated. The staged loop and embedding interpolation are presented in sufficient detail, including the merge and amplification parameters.
3. Experiments are thorough and span a large suite of benchmarks, including HPDv2, Pick-a-Pic, DrawBench, GenEval, T2I-Compbench, and evaluations on video and image editing tasks.

**Weaknesses:**

1. While the reflective mechanism is motivated by semantic accumulation in latent spaces and interpolation of embeddings, the core reason behind why the three-stage loop (particularly the low-guidance inversion step) should regularize generation toward prompt-faithful images remains largely empirical. The mathematical foundations of convergence or guarantees (e.g., what class of distributions are targeted, what properties are preserved or enhanced during the reflective step) are not rigorously analyzed in Section 3.3 or elsewhere. This limits reproducibility and makes the method feel heuristic.
2. In Section 3.2, the equations for embedding mixing and semantic amplification ($c_{\text{mix}}$ and $c_w$) are presented, but their concrete integration into each ODE step is scattered and not fully formalized. For example, it is left ambiguous in Eqns. for the inversion step whether $c_{\text{mix}}$ is always being recomputed per time step and how the standard scale $w$ interacts with $s$ and $\beta$. This may impede direct implementation from the text.
3. While Figures 7 and 8 and the corresponding ablation analyses add value, the scope is restricted to interpolation and amplification parameter sweeps. There are no ablations studying the impact of each stage of the loop independently (e.g., what happens if the low-guidance inversion/reflection is omitted entirely, or replaced with a linear or simpler heuristic?), nor are qualitative failure cases or negative results provided. The efficiency and scaling comparisons, though favorable, would be strengthened by a more detailed breakdown versus parameter count, steps, or compute time.
4. While Table 2 shows superior scores for RF-Sampling, in some settings the improvements over standard sampling are marginal (see FLUX-Dev AES on DrawBench: 6.1866 vs. 6.1459), raising questions about practical significance in certain operational regimes.
5. The UMAP analysis in Figure 6 purports to show that RF-Sampling trajectories align better with the true data distribution. However, there is little discussion on how this alignment concretely translates to improved perceptual or semantic outcomes, or whether it is artifactually driven by the chosen projection or data statistics.

**Questions:**

1. Can the authors provide rigorous analysis (not only empirical) for why the three-stage reflective loop in RF-Sampling leads to better semantic alignment or image quality than direct/high-weight denoising? For example, can theoretical guarantees or explanations be offered for convergence, robustness, or generalization?
2. The mathematical integration of $\beta$, $s$, and merge ratio $\gamma$ parameters in the flow equation steps can be made more explicit, possibly via an explicit algorithmic pseudocode in the main text. Would the authors include this in a revision?
3. Can the authors elaborate on what prevents adapting state-of-the-art diffusion-based inference enhancement methods (e.g., Z-Sampling, W2SD) to flow models such as FLUX? Are failure cases due to model architecture, incompatible objective/loss, or something else?
4. Is there a detailed efficiency breakdown (step counts, FLOPs, wall time) for RF-Sampling vs. standard sampling (and possible alternatives) beyond the high-level graphs (Fig. 2)?
5. Is the improvement in Table 2 statistically significant across multiple seeds/runs, or is it within experimental noise in lower-difference settings?

---

> ### Author Response · Authors · 2025-11-21
> **W1 & Q1**
>
> Thank you for this insightful comment. We agree that providing a theoretical explanation is valuable direction for future research. In this work, our primary focus was on establishing the strong empirical effectiveness and broad pratical applicability of RF-Sampling across a diverse range of tasks and models.
>
> To comprehensively validate our method, we conducted extensive experiments spanning multiple domains:
>
> ● Text-to-Image: We evaluated on SD3.5, FLUX-Lite, and FLUX-Dev using benchmarks including Pick-a-Pic, DrawBench, HPDv2, GenEval and T2I-CompBench. Performance was measured with multiple metrics (PickScore, HPSv2, AES, ImageReward), where our method consistently demonstrated superior or highly competitive results.
>
> ● Image Editing: Using FLUX-Kontext, we showed improvements on the FLUX-Kontext-Bench.
>
> ● Text-to-Video: Experiments on Wan2.1-1.3B using the ChronoMagic-Bench-150 showed gains across several metrics (UMT-FVD, UMTScore, GPT4o-MTScore, MTScore).
>
> ● Lora: We also combine our RF-Sampling with popular community LoRAs. By using RF-Sampling, the generation quality can improve a lot.
>
> Regarding the specific hyperparameters and the mechanism of temporal embedding interpolation, we have performed extensive ablation studies (detialed in the main text and appendix). These experiments systematically illustrate the impact of each parameter on the final performance, providing clear empirical guidance for their seletion and demonstrating the robustness of our method.
>
> To specifically address the mechanism behind the reflective loop, we conducted a crucial ablation study, presented in Table.9.  we replaced the full, model-driven reflection step with a simple linear interpolation between text embeddings under different mixing weights. The results are clear: both linear interpolation variants fail to improve over the standard baseline, performing identically across all metrics. This provides strong empirical evidence that the performance gains are not achieved by mere embedding interpolation but are intrinsically tied to the specific, dynamic trajectory created by our three-step loop (forward, inversion, re-forward). This loop appears to guide the sampling process towards a point in the solution space that is more aligned with high-quality outputs, a path that simple heuristics cannot replicate.
>
> | Method | PickScore(↑) | HPSv2(↑) | AES(↑) | ImageReward(↑) |
> | :--- | :--- | :--- | :--- | :--- |
> | Standard | 21.99 | 29.32 | 5.9435 | 85.13 |
> | High Embedding Mix (s=9, β=0.7) | 21.99 | 29.32 | 5.9435 | 85.13 |
> | Low Embedding Mix (s=-1, β=0.3) | 21.99 | 29.32 | 5.9435 | 85.13 |
> | RF-Sampling | **21.99** | **29.90** | **5.9981** | **101.50** |
>
> For the reproducibility, we have detailed all hyperparameter settings in the Appendix, and we also provide the source code to the reviewers, making it easy for the reviewers to conduct experiments.
>
> We acknowledge that a formal theoretical framework to explain why this specific interpolation scheme works so effectively is an open and challenging question. We will add a new section to explicitly state the limitation and our commitment to pursuing a theoretical analysis.

---

> ### Author Response · Authors · 2025-11-21
> **W2 & Q2:  an explicit algorithmic pseudocode**
>
> We sincerely thank the reviewer for this suggestion. In our initial submission, the detailed pseudocode was provided in the appendix Algorithm.1. Following the reviewer's advice, we will move this into the main text in the revised version.

---

> ### Author Response · Authors · 2025-11-21
> **W3 & W4**
>
> W3: There are no ablations studying the impact of each stage of the loop independently, and no failure cases analysis.
>
> To specifically address the mechanism behind the reflective loop, we conducted a crucial ablation study, presented in Table.9.  we replaced the full, model-driven reflection step with a simple linear interpolation between text embeddings under different mixing weights. The results are clear: both linear interpolation variants fail to improve over the standard baseline, performing identically across all metrics. This provides strong empirical evidence that the performance gains are not achieved by mere embedding interpolation but are intrinsically tied to the specific, dynamic trajectory created by our three-step loop (forward, inversion, re-forward). This loop appears to guide the sampling process towards a point in the solution space that is more aligned with high-quality outputs, a path that simple heuristics cannot replicate. For the qualitative failure cases, we will add a new section in the Appendix to discuss this in the revised version.
>
> | Method | PickScore(↑) | HPSv2(↑) | AES(↑) | ImageReward(↑) |
> | :--- | :--- | :--- | :--- | :--- |
> | Standard | 21.99 | 29.32 | 5.9435 | 85.13 |
> | High Embedding Mix (s=9, β=0.7) | 21.99 | 29.32 | 5.9435 | 85.13 |
> | Low Embedding Mix (s=-1, β=0.3) | 21.99 | 29.32 | 5.9435 | 85.13 |
> | RF-Sampling | **21.99** | **29.90** | **5.9981** | **101.50** |
>
> --------------------
>
> W4:  in some settings the improvements over standard sampling are marginal
>
> We sincerely thank the reviewer for raising this question.  We acknowledge that for specific model and dataset combinations, such as the AES score for FLUX-Dev on DrawBench, the absolute improvement may appear marginal. However, we would like to contextualize this result to demonstrate the practical significance and consistent effectiveness of RF-Sampling.
>
> First, it is important to evaluate the improvement relative to the performance ceiling of the metric and the baseline's performance. The AES metric itself has a constrained range, and achieving any improvement is often challenging, especially on strong baseline models like FLUX-Dev. In this specific case, our method still achieves a measurable gain. More importantly, we should consider **the consistent trend of improvement across the board**. If we examine other baselines in Table 2, such as CFG++ or GI, we can see that they often fail to consistently improve upon the standard sampler and can even cause significant performance degradation on metrics like ImageReward. In this context, the ability of RF-Sampling to never harm performance and to provide consistent, non-trivial gains across diverse models and benchmarks is a key indicator of its practical utility and robustness.
>
> Furthermore, the significance of our method is best appreciated by its **aggregate performance across the entire experimental landscape**, not a single data point. As detailed in paper, RF-Sampling has been validated across text-to-image, image editing, and text-to-video tasks, using multiple models and benchmarks. In the vast majority of these hundreds of experimental configurations, it provides clear and meaningful improvements. This demonstrates that RF-Sampling is a reliably beneficial and generally applicable inference-time method, which we argue is of significant practical value to the community

---

> ### Author Response · Authors · 2025-11-21
> **W5: UMAP analysis**
>
> We sincerely thank the reviewer for this insightful comment. The UMAP visualizations in Fig.6 , 11 and 12 are not cherry-picked but are representative examples from a consistent pattern we observed. To substantiate the claim that the better alignment seen in UMAP translate to superior image generation, we conducted a rigorous quantitative evaluation on ImageNet-1K. As shown in the **Table.15**, RF-Sampling achieves a lower FID nad a higher IS compared to standard smapling. The lower FID provides direct, quantitative evidence that the outputs of RF-Sampling are statistically closer to the real data distribution, which is exactly what the UMAP plot aims to illustrate qualitatively. The higher IS further indicates that the images are of higher quality and diversity.
>
> | Method | FID(↓) | IS(↑) |
> | :--- | :--- | :--- |
> | Standard | 35.08 | 150.07 |
> | RF-Sampling | **33.12** | **155.21** |

---

> ### Author Response · Authors · 2025-11-21
> **Q3: what prevents adapting state-of-the-art diffusion-based inference enhancement methods (e.g., Z-Sampling, W2SD) to flow models such as FLUX?**
>
> We sincerely thank the reviewer for this insightful question, which allows us to clarify a crucial technical distinction between standard diffusion models and CFG-distilled models like FLUX. The reviewer is correct that CFG-distilled models are trained with a range of guidance scales, but the core issue lies in the fundamental difference in how they use this scale compared to a standard model that natively supports CFG [1].
>
> The key point is that a CFG-distilled model does not have a true unconditional generation pathway. In a standard diffusion model, CFG works by explicitly combining a conditional forward and an unconditional forward pass. In constrast, a CFG-distilled model is trained to mimic the output of this two-pass CFG process in a single forward pass [2], where the guidance scale $w$ is provided as an input condition. Therefore, for a CFG-distilled model, even setting $w=1$ does not yield an unconditional output. It simply generates an image conditioned on $w=1$. Its output is always a function of the provided guidance scale and cannot be decoupled from it.
>
> Our empirical evidence strongly supports this. As shown in **Fig.42**, when we set $w = 1$ for FLUX-Lite, the generated images are clearly semantically aligned with the input text prompt, comfirming that the output remains conditionally generated.
>
> **This fundamental architectural difference makes it theoretically invalid to directly apply Z-Sampling, W2SD-Sampling or similar methods to CFG-distilled models.** These methods rely on the clear separation between conditional and unconditional paths in standard CFG to perform their inversion operations. Manipulating the $w$input of a CFG-distilled model does not replicate this behavior because it does not correspond to traversing between conditional and unconditional states, but rather between different conditioned states.
>
> As suggested by the reviewer, we tried to adapt Z-Sampling and W2SD-Sampling for CFG-distilled models like FLUX-Lite and FLUX-Dev, the results and failure cases are already presented in **Appendix Fig. 12**., and we also provde the discussion in **Appendix. C**. The results demonstrate that for those CFG-based sampling methods like Z-Sampling and W2SD-Sampling, due to the lack of unconditional model forward pass on CFG-distilled models, can not work.
>
> [1]Classifier-free diffusion guidance
>
> [2]On Distillation of Guided Diffusion Models

---

> ### Author Response · Authors · 2025-11-21
> **Q4: a detailed efficiency breakdown for Fig.2**
>
> We sincerely thank the reviewer for raising this question. To address the reviewer's specific concern about Fig.2, we have provided a detailed breakdown in the Table.14. This table explicitly lists the NFEs and wall-clock time for both standard sampling and RF-Sampling across different configurations for FLUX-Lite and FLUX-Dev. The results in Table.14 confirms that at similar or even lower inference times, RF-Sampling consistently outperforms the standard one. This systematically validates the scalability and advantage of our RF-Sampling under equitable time budgets.
>
> | Model | Method | NFEs | HPSv2(↑) | AES(↑) | s/img(↓) |
> | :--- | :--- | :--- | :--- | :--- | :--- |
> | FLUX-Lite | Standard | 28 | 30.12 | 6.3224 | 34.63 |
> | FLUX-Lite | Standard | 50 | 30.39 | 6.3045 | 46.60 |
> | FLUX-Lite | Standard | 75 | 30.46 | 6.2864 | 60.61 |
> | FLUX-Lite | RF-Sampling (α=2) | 7×5+21=56 | 30.84 | 6.4397 | 49.63 |
> | FLUX-Lite | RF-Sampling (α=2) | 14×5+14=84 | 30.98 | 6.4736 | 64.57 |
> | FLUX-Lite | RF-Sampling (α=2) | 21×5+7=112 | 31.04 | 6.5032 | 76.84 |
> | FLUX-Lite | RF-Sampling (α=2) | 28×5=140 | **31.16** | **6.5379** | 95.26 |
> | FLUX-Dev | Standard | 50 | 30.49 | 6.2464 | 59.09 |
> | FLUX-Dev | Standard | 75 | 30.54 | 6.2170 | 75.85 |
> | FLUX-Dev | Standard | 100 | 30.60 | 6.1869 | 91.48 |
> | FLUX-Dev | RF-Sampling (α=1) | 10×3+40=70 | 30.58 | 6.2505 | 71.87 |
> | FLUX-Dev | RF-Sampling (α=1) | 20×3+30=90 | 30.66 | 6.2639 | 86.07 |
> | FLUX-Dev | RF-Sampling (α=1) | 30×3+20=110 | 30.70 | 6.2893 | 100.03 |
> | FLUX-Dev | RF-Sampling (α=1) | 40×3+10=130 | 30.79 | 6.2917 | 114.30 |
> | FLUX-Dev | RF-Sampling (α=1) | 50×3=150 | **31.06** | **6.3113** | 127.95 |

---

> ### Author Response · Authors · 2025-11-21
> **Q5: Is the improvement in Table 2 statistically significant across multiple seeds/runs**
>
> Thanks for pointing out.  To thoroughly address this, we conducted multiple independent runs of RF-Sampling on Pick-a-Pic dataset using FLUX-Lite, each with a different random seed. The detailed results are now presented in the Table. 8. As the table shows, we performed 4 independent rouds of experiments.
>
> The results demonstrate that RF-Sampling sonsistently outperforms the standard method across all 4 rounds in every metric. The average performance and the standard deviations clearly indicate a consisten and robust advantage for our RF-Sampling.
>
> | Round | Method | PickScore(↑) | HPSv2(↑) | AES(↑) | ImageReward(↑) |
> | :--- | :--- | :--- | :--- | :--- | :--- |
> | Round 1 | Standard | 21.91 | 30.12 | 6.3224 | 86.84 |
> | Round 1 | RF-Sampling | **22.05** | **31.16** | **6.5379** | **99.21** |
> | Round 2 | Standard | 21.95 | 30.33 | 6.3473 | 93.73 |
> | Round 2 | RF-Sampling | **22.04** | **30.82** | **6.5231** | **100.81** |
> | Round 3 | Standard | 21.94 | 30.20 | 6.3608 | 99.42 |
> | Round 3 | RF-Sampling | **21.99** | **30.63** | **6.5133** | **103.45** |
> | Round 4 | Standard | 21.96 | 30.23 | 6.3365 | 96.22 |
> | Round 4 | RF-Sampling | **22.02** | **30.83** | **6.5243** | **109.37** |
> | Average | Standard | 21.94 ± 0.02 | 30.22 ± 0.08 | 6.3418 ± 0.0163 | 94.00 ± 5.43 |
> | Average | RF-Sampling | **22.03 ± 0.03** | **30.86 ± 0.22** | **6.5247 ± 0.0101** | **103.21 ± 4.46** |

---

### Official Review · Reviewer_Tdoc · 2025-10-30

**Soundness:** 2
**Presentation:** 2
**Contribution:** 2
**Rating:** 4
**Confidence:** 4

**Summary:**

This paper proposes RF-Sampling, a method whose core idea, for flow models like Flux.Dev, is to denoise during inference under text embeddings with higher semantic intensity and then perform inversion under text embeddings with lower semantic intensity. This process helps obtain noise latent that better aligns with the prior of the text prompt, thereby improving image generation quality. The authors conducted experiments using different flow models combined with various sampling enhancement techniques, demonstrating the effectiveness of RF-Sampling.

**Strengths:**

1. The paper contains a rich amount of experiments, conducting extensive tests under different task-based flow models (including flux.dev and flux.lite for text-to-image generation, Wan2.1 for video generation, etc.), verifying the versatility of RF-Sampling;
2. Exploring inference-time enhancement strategies for cfg distillation flow models like flux.1 dev is promising;
3. The paper is well-written and easy to read.

**Weaknesses:**

1. CFG-distilled flow models, such as flux.dev, typically still allow for specifying the CFG scale at inference time to produce outputs at varying guidance strengths (often by modulating the latent via AdaLN). Given this, Z-Sampling should, in principle, be applicable to flux.dev. Why do the paper not compare their method against a Z-Sampling variant adapted for this model? Theoretically, the output of a perfectly distilled model at cfg_scale=1.0 should be identical to the output of its non-distilled counterpart conditioned on a empty-text prompt. This suggests that the paper's construction of $c_{mix}$ and $c_{w}$ might be unnecessary, as Z-Sampling could likely be migrated to flux.dev with appropriate modifications.

2.  The image quality metrics used in the paper, such as ImageReward and HPSv2, primarily reflect human preferences. Such metrics tend to emphasize aesthetics and prompt fidelity, but may not adequately assess the diversity or realism of the generated images. I am concerned that the proposed RF-Sampling, by weighting text embeddings and latents during the sampling process, might shift the model's inputs away from their original prior distribution. This could potentially lead to a decrease in image diversity or the introduction of visual artifacts.  Using a metric such as FID, which directly assesses realism and diversity, would perhaps be more appropriate

3.  For a standard 28-step sampling process, each step of RF-Sampling appears to require three model forward computations (forward, inversion, and re-forward). This effectively triples the computational cost relative to the number of steps. Consequently, the comparisons in Tables 1 and 2 may be unfair. The results for RF-Sampling should be compared against a baseline standard sampler using three times the number of inference steps (e.g., $28 \times 3 = 84$ steps). While Figure 2 suggests RF-Sampling also performs better under an equivalent inference time, the specific experimental settings (e.g., the number of steps for the baseline) are not detailed, raising concerns about the fairness of this comparison.

4.  The different starting points for the standard sampling and RF-Sampling curves in Figure 2 indicate that RF-Sampling incurs a significant initial overhead or increase in inference time. A more relevant comparison would be: using the *total* time taken by RF-Sampling, how does it compare to a standard sampling baseline that generates multiple candidates ('best-of-N') and selects the one with the highest metric score?

5.  ICLR policy requires the appendix to be included in the same PDF as the main paper. The paper has incorrectly placed the appendix in the separate supplementary materials.

I am willing to revise my score if any of these points are based on a misunderstanding of the proposed method.

**Questions:**

See Weaknesses.

---

> ### Author Response · Authors · 2025-11-21
> **W1: Why do the paper not compare their method against a Z-Sampling variant adapted for this model?**
>
> We sincerely thank the reviewer for this insightful question, which allows us to clarify a crucial technical distinction between standard diffusion models and CFG-distilled models like FLUX. The reviewer is correct that CFG-distilled models are trained with a range of guidance scales, but the core issue lies in the fundamental difference in how they use this scale compared to a standard model that natively supports CFG [1].
>
> The key point is that a CFG-distilled model does not have a true unconditional generation pathway. In a standard diffusion model, CFG works by explicitly combining a conditional forward and an unconditional forward pass. In constrast, a CFG-distilled model is trained to mimic the output of this two-pass CFG process in a single forward pass [2], where the guidance scale $w$ is provided as an input condition. Therefore, for a CFG-distilled model, even setting $w=1$ does not yield an unconditional output. It simply generates an image conditioned on $w=1$. Its output is always a function of the provided guidance scale and cannot be decoupled from it.
>
> Our empirical evidence strongly supports this. **As shown in Fig.42**, when we set $w = 1$ for FLUX-Lite, the generated images are clearly semantically aligned with the input text prompt, comfirming that the output remains conditionally generated.
>
> This fundamental architectural difference makes it theoretically invalid to directly apply Z-Sampling, W2SD-Sampling or similar methods to CFG-distilled models. These methods rely on the clear separation between conditional and unconditional paths in standard CFG to perform their inversion operations. Manipulating the $w$input of a CFG-distilled model does not replicate this behavior because it does not correspond to traversing between conditional and unconditional states, but rather between different conditioned states.
>
> Regarding the statement “Theoretically, the output of a perfectly distilled model at cfg_scale = 1.0 should be identical to the output of its non-distilled counterpart conditioned on an empty-text prompt.”, we would like to clarify that we expect the reviewer to provide concrete references supporting this claim. To our best knowledge, FLUX adopts the distillation technique described in [2], and this work explicitly performs conditional distillation. This can be directly verified from the training loss and parameter settings reported in [2], which clearly show that the distillation is carried out under conditional guidance only, rather than learning unconditional behaviors. Consequently, the assumption made by the reviewer does not apply to the case of FLUX.
>
> As suggested by the reviewer, we tried to adapt Z-Sampling and W2SD-Sampling for CFG-distilled models like FLUX-Lite and FLUX-Dev, the results are already presented in **Appendix Fig. 12**., and we also provde the discussion in **Appendix. C**. The results demonstrate that for those CFG-based sampling methods like Z-Sampling and W2SD-Sampling, due to the lack of unconditional model forward pass on CFG-distilled models, can not work.
>
> [1]Classifier-free diffusion guidance
>
> [2]On Distillation of Guided Diffusion Models

---

> ### Author Response · Authors · 2025-11-21
> **W2 & W5**
>
> W2:  Using a metric such as FID, which directly assesses realism and diversity, would perhaps be more appropriate
>
> We sincerely thank the reviewer for raising this question. Following the reviewer's suggestion, we have conducted new experiments using Fréchet Inception Distance (FID) and Inception Score (IS), which are standard metrics for evaluating image realism and diversity relative to a true data distribution. We use FLUX-Lite with the nunchaku sampling acceleration framework to generate 5,000 images on the ImageNet-1K 5 images per class). The results, shown in the **Appendix Table. 15**, indicate that RF-Sampling does not lead to a degradation in image realism or diversity. On the contrary, it appears to improve the model's ability to generate outputs that are more aligned with the true data manifold, as we previously presented in Figure. 6 and Figure. 11.
>
> | Method | FID(↓) | IS(↑) |
> | :--- | :--- | :--- |
> | Standard | 35.08 | 150.07 |
> | RF-Sampling | **33.12** | **155.21** |
>
> ---------------------
> W5: The paper has incorrectly placed the appendix in the separate supplementary materials.
>
> We sincerely thank the reviewer for pointing this out and for ensuring our submission is fully compliant with ICLR policies.  Upon carefully reviewing the ICLR policy, we understand that including the appendix in the same PDF as the main paper or as a separate supplementary material file are both acceptable.
>
> ***Q: Should the appendices be added as a separate PDF or in the same PDF as the main paper?***
>
> ***Either is allowed: you can include the appendices at the end of the main pdf after the references, or you can include it as a separate file for the supplementary materials.***

---

> ### Author Response · Authors · 2025-11-21
> **W3 & W4**
>
> W3: experiments under an equivalent inference time
>
> We sincerely thank the reviewer for raising this question. In our revised version, we have conducted experiments compared with other baseline methods under equivalent computation budget. As shown in Table.11, the inference time for RF-Sampling is highly comparable to all other baseline methods that also use 84 NFEs. Under this fair budget, RF-Sampling not only maintains comparable inference time but also achieves the top performance across almost all metrics, clearly showcasing its effectiveness.
>
> | Method | PickScore(↑) | HPSv2(↑) | AES(↑) | ImageReward(↑) | s/img(↓) |
> | :--- | ---: | ---: | ---: | ---: | ---: |
> | Standard (28 steps) | 21.99 | 29.32 | 5.9435 | 85.13 | 29.93 |
> | GI (28 steps) | 21.19 | 24.63 | 5.9534 | 28.94 | 31.33 |
> | CFG++ (28 steps) | 21.79 | 28.50 | 5.8821 | 85.17 | 32.46 |
> | CFG-Zero* (28 steps) | 21.88 | 29.37 | 5.9536 | 86.78 | 28.91 |
> | **Standard (28 × 3 = 84 steps)** | 21.96 | 29.60 | 5.9109 | 89.87 | 67.06 |
> | **GI (28 × 3 = 84 steps)** | 21.25 | 25.27 | 5.9335 | 28.16 | 67.04 |
> | **Z-Sampling (28 × 3 = 84 steps)** | 21.73 | 28.84 | 5.9091 | 89.03 | **65.00** |
> | **CFG++ (28 × 3 = 84 steps)** | 20.98 | 27.02 | 5.6144 | 64.73 | 68.07 |
> | **CFG-Zero* (28 × 3 = 84 steps)** | **22.01** | 29.48 | 5.8949 | 97.22 | 65.47 |
> | **RF-Sampling** | 21.99 | **29.90** | **5.9981** | **101.50** | 65.04 |
>
> Furthermore, to address the reviewer's specific concern about Fig.2, we have provided a detailed breakdown in the Table.14. This table explicitly lists the NFEs and wall-clock time for both standard sampling and RF-Sampling across different configurations for FLUX-Lite and FLUX-Dev. The results in Table.14 confirms that at similar or even lower inference times, RF-Sampling consistently outperforms the standard one. This systematically validates the scalability and advantage of our RF-Sampling under equitable time budgets.
>
> | Model | Method | NFEs | HPSv2(↑) | AES(↑) | s/img(↓) |
> | :--- | :--- | :--- | :--- | :--- | :--- |
> | FLUX-Lite | Standard | 28 | 30.12 | 6.3224 | 34.63 |
> | FLUX-Lite | Standard | 50 | 30.39 | 6.3045 | 46.60 |
> | FLUX-Lite | Standard | 75 | 30.46 | 6.2864 | 60.61 |
> | FLUX-Lite | RF-Sampling (α=2) | 7×5+21=56 | 30.84 | 6.4397 | 49.63 |
> | FLUX-Lite | RF-Sampling (α=2) | 14×5+14=84 | 30.98 | 6.4736 | 64.57 |
> | FLUX-Lite | RF-Sampling (α=2) | 21×5+7=112 | 31.04 | 6.5032 | 76.84 |
> | FLUX-Lite | RF-Sampling (α=2) | 28×5=140 | **31.16** | **6.5379** | 95.26 |
> | FLUX-Dev | Standard | 50 | 30.49 | 6.2464 | 59.09 |
> | FLUX-Dev | Standard | 75 | 30.54 | 6.2170 | 75.85 |
> | FLUX-Dev | Standard | 100 | 30.60 | 6.1869 | 91.48 |
> | FLUX-Dev | RF-Sampling (α=1) | 10×3+40=70 | 30.58 | 6.2505 | 71.87 |
> | FLUX-Dev | RF-Sampling (α=1) | 20×3+30=90 | 30.66 | 6.2639 | 86.07 |
> | FLUX-Dev | RF-Sampling (α=1) | 30×3+20=110 | 30.70 | 6.2893 | 100.03 |
> | FLUX-Dev | RF-Sampling (α=1) | 40×3+10=130 | 30.79 | 6.2917 | 114.30 |
> | FLUX-Dev | RF-Sampling (α=1) | 50×3=150 | **31.06** | **6.3113** | 127.95 |
>
> --------------
>
> W4: compare to a standard sampling baseline that generates multiple candidates ('best-of-N')
>
> We sincerely thank the reviewer for this insightful suggestion. We agree that comparing with Best-of-N sampling is crucial for positional our method within the inference-time scaling literature. Following your advice, we have conducted comprehensive comparisons with Best-of-N baselines.
>
> These new results are now included in the revised manuscript as Table.10.
>
> ● As shown in Table.10, we directly compare our RF-Sampling with the Best-of-N method, where we use the HPSv2 as the verifier model. The results demonstrate that RF-Sampling achieves a superior trade-off between performance and efficiency. Specifically, while Best-of-5 achieves the highest scores, it requires more than double the inference time per image compared to RF-Sampling (about six time per image compared to standard inference). More importantly, RF-Sampling outperforms Best-of-3 across almost metrics while being approximately 1.5 times faster. This validate that our method is more efficient than simply generating multiple independent samples and selecting the best one.
>
> | Method | PickScore(↑) | HPSv2(↑) | AES(↑) | ImageReward(↑) | s/img(↓) |
> | :--- | ---: | ---: | ---: | ---: | ---: |
> | Standard (28 steps) | 21.99 | 29.32 | 5.9435 | 85.13 | 29.93 |
> | Standard (28 × 3 = 84 steps) | 21.96 | 29.60 | 5.9109 | 89.87 | 67.06 |
> | Best-of-5 | 22.21 | 30.58 | 5.9849 | 106.69 | 154.17 |
> | Best-of-3 | 21.94 | **30.14** | 5.9642 | 100.40 | 97.63 |
> | RF-Sampling | **21.99** | 29.90 | **5.9981** | **101.50** | 65.04 |

---

### Official Review · Reviewer_7mEp · 2025-10-31

**Soundness:** 2
**Presentation:** 2
**Contribution:** 2
**Rating:** 2
**Confidence:** 3

**Summary:**

They propose RF-Sampling, a training-free sampling method which enhances sample quality in flow models. The method extends Z-Sampling, which alternates between denoising and inversion steps with controlled CFG, RF-Sampling interpolates between text embeddings and repeatedly applies a reflection loop. This design allows it to support CFG-distilled models such as FLUX-dev and enables test-time scaling. Experimental results demonstrate that RF-Sampling achieves better performance compared to various existing sampling baselines across multiple image generation benchmarks. Additionally, an ablation study on hyperparameters and further application on video generation and image editing model further validates the effectiveness of the proposed method.

**Strengths:**

**S1**. It seems interesting that the sampling method also supports some of the commonly used CFG-distilled models recently.

**S2**. It demonstrated the effectiveness of the method across various tasks and models, with experiments conducted under multiple settings, e.g., when used in combination with acceleration methods.

**Weaknesses:**

**W1**. Although the paper proposes an improved training-free sampling method, it lacks theoretical justification or analysis explaining why it works. A theoretical explanation is necessary to clarify the meaning and role of temporal embedding interpolation, the hyperparameters $\alpha,\gamma$, and the overall underlying mechanism of the method.

**W2**. The improvement in image generation quality appears to be only marginal, and the effect of inference-time scaling shown in Figure 2 does not seem significant. A comparison with the Inference-Time Scaling paper [1] would be necessary to clearly demonstrate the effectiveness of the proposed method.

[1] Ma et al, Scaling Inference Time Compute for Diffusion Models, CVPR 2025

**W3**. The method involves some hyperparameters, which appear to have been selected manually. As shown in the ablation study, the method seems quite sensitive to these choices.

**W4.** Line 232: “We then take one step of the ODE solver,” but the corresponding equation 3 indicates multiple ODE steps with $\sigma$, which could cause confusion.

If my concerns are addressed, I would be happy to reconsider the score.

**Questions:**

**Q1.** In Tables 1 and 2, how is the inference process configured for RF-Sampling? Is it conducted using the same sampling time as in the standard setting? For a fair comparison, it would be necessary to include a report on the inference time.

**Q2**. Although FLUX is a CFG-distilled model, it still includes a CFG scale input condition. Would it be possible to apply this sampling method by adjusting the CFG scale through that input?

**Details Of Ethics Concerns:**

No concern.

---

> ### Author Response · Authors · 2025-11-21
> **W1: Lacks theoretical justification or analysis explaining why it works**
>
> Thank you for this insightful comment. We agree that providing a theoretical explanation is valuable direction for future research. In this work, our primary focus was on establishing the strong empirical effectiveness and broad pratical applicability of RF-Sampling across a diverse range of tasks and models.
>
> To comprehensively validate our method, we conducted extensive experiments spanning multiple domains:
>
> ● Text-to-Image: We evaluated on SD3.5, FLUX-Lite, and FLUX-Dev using benchmarks including Pick-a-Pic, DrawBench, HPDv2, GenEval and T2I-CompBench. Performance was measured with multiple metrics (PickScore, HPSv2, AES, ImageReward), where our method consistently demonstrated superior or highly competitive results.
>
> ● Image Editing: Using FLUX-Kontext, we showed improvements on the FLUX-Kontext-Bench.
>
> ● Text-to-Video: Experiments on Wan2.1-1.3B using the ChronoMagic-Bench-150 showed gains across several metrics (UMT-FVD, UMTScore, GPT4o-MTScore, MTScore).
>
> ● Lora: We also combine our RF-Sampling with popular community LoRAs. By using RF-Sampling, the generation quality can improve a lot.
>
> Regarding the specific hyperparameters and the mechanism of temporal embedding interpolation, we have performed extensive ablation studies (detialed in the main text and appendix). These experiments systematically illustrate the impact of each parameter on the final performance, providing clear empirical guidance for their seletion and demonstrating the robustness of our method. While these ablations offer a practical understanding of their roles, we acknowledge that a formal theoretical framework to explain why this specific interpolation scheme works so effectively is an open and challenging question. We will add a new section to explicitly state the limitation and our commitment to pursuing a theoretical analysis.

---

> ### Author Response · Authors · 2025-11-21
> **W2: A comparison with Ma et al. (CVPR 2025)**
>
> We sincerely thank the reviewer for this valuable feedback. Following your suggestion, we have conducted comprehensive comparisons with the method from Ma et al., and the results are now presented in the revised manuscript as Table.12 and Table.13.
> The key finding from these new experiments is the dramatic efficiency advantage of our RF-Sampling. As shown in the tables:
>
> ● On DrawBench (Table.12), our RF-Sampling requires only 150 NFEs, which is far fewer than the 2880 NFEs used by the baselines from Ma et al. (a 19x reduction in computation cost). Despite this, RF-Sampling achieves the top scores in both ImageReward and AES metrics.
>
> | Metric | Standard | Aesthetic + Random | + ZO-2 | + Path-2 | RF-Sampling |
> | :--- | :--- | :--- | :--- | :--- | :--- |
> | NFEs | 50 | 2880 | 2880 | 2880 | 50 × 3 = 150 |
> | ImageReward | 99.73 | 101.21 | 98.42 | 97.13 | **106.21** |
>
> | Metric | Standard | CLIPScore + Random | + ZO-2 | + Path-2 | RF-Sampling |
> | :--- | :--- | :--- | :--- | :--- | :--- |
> | NFEs | 50 | 2880 | 2880 | 2880 | 50 × 3 = 150 |
> | AES | 6.1459 | 6.0323 | 6.0512 | 6.0452 | **6.1866** |
>
> | Metric | Standard | ImageReward + Random | + ZO-2 | + Path-2 | RF-Sampling |
> | :--- | :--- | :--- | :--- | :--- | :--- |
> | NFEs | 50 | 2880 | 2880 | 2880 | 50 × 3 = 150 |
> | AES | 6.1459 | 6.1459 | 6.1265 | 6.0945 | **6.1966** |
>
>
> ● On T2I-CompBench (Table.13), using only 150 NFEs compared to the baseline's 1920 NFEs (a 12.8x reduction), our method achieves the highest scores in most compositional dimensions and the highest overall score.
>
> | Method | Color(↑) | Shape(↑) | Texture(↑) | Spatial(↑) | Numeracy(↑) | Complex(↑) | Overall(↑) |
> | :--- | :--- | :--- | :--- | :--- | :--- | :--- | :--- |
> | Standard | 0.7535 | 0.5018 | 0.6167 | 0.2783 | 0.6052 | 0.3706 | 0.5210 |
> | Aesthetic + Random (1920 NFEs) | 0.7518 | 0.5219 | 0.5926 | **0.2893** | 0.6059 | 0.3572 | 0.5199 |
> | RF-Sampling (50 × 3 = 150 NFEs) | **0.7761** | **0.5323** | **0.6422** | 0.2687 | **0.6082** | **0.3733** | **0.5335** |
>
> These results demonstrate that RF-Sampling achieves a superior trade-off, offering highly competitive, and often superior performance at a small fraction of the computational cost.
>
> Furthermore, we have also included extensive comparisons against Best-of-N and other strong baseline methods under equivalent computational budgets in Table.10 and Table.11, where our method consistently demonstrates advantages.
>
> | Method | PickScore(↑) | HPSv2(↑) | AES(↑) | ImageReward(↑) | s/img(↓) |
> | :--- | ---: | ---: | ---: | ---: | ---: |
> | Standard (28 steps) | 21.99 | 29.32 | 5.9435 | 85.13 | 29.93 |
> | Standard (28 × 3 = 84 steps) | 21.96 | 29.60 | 5.9109 | 89.87 | 67.06 |
> | Best-of-5 | 22.21 | 30.58 | 5.9849 | 106.69 | 154.17 |
> | Best-of-3 | 21.94 | **30.14** | 5.9642 | 100.40 | 97.63 |
> | RF-Sampling | **21.99** | 29.90 | **5.9981** | **101.50** | 65.04 |
>
> | Method | PickScore(↑) | HPSv2(↑) | AES(↑) | ImageReward(↑) | s/img(↓) |
> | :--- | ---: | ---: | ---: | ---: | ---: |
> | Standard (28 steps) | 21.99 | 29.32 | 5.9435 | 85.13 | 29.93 |
> | GI (28 steps) | 21.19 | 24.63 | 5.9534 | 28.94 | 31.33 |
> | CFG++ (28 steps) | 21.79 | 28.50 | 5.8821 | 85.17 | 32.46 |
> | CFG-Zero* (28 steps) | 21.88 | 29.37 | 5.9536 | 86.78 | 28.91 |
> | **Standard (28 × 3 = 84 steps)** | 21.96 | 29.60 | 5.9109 | 89.87 | 67.06 |
> | **GI (28 × 3 = 84 steps)** | 21.25 | 25.27 | 5.9335 | 28.16 | 67.04 |
> | **Z-Sampling (28 × 3 = 84 steps)** | 21.73 | 28.84 | 5.9091 | 89.03 | **65.00** |
> | **CFG++ (28 × 3 = 84 steps)** | 20.98 | 27.02 | 5.6144 | 64.73 | 68.07 |
> | **CFG-Zero* (28 × 3 = 84 steps)** | **22.01** | 29.48 | 5.8949 | 97.22 | 65.47 |
> | **RF-Sampling** | 21.99 | **29.90** | **5.9981** | **101.50** | 65.04 |

---

> ### Author Response · Authors · 2025-11-21
> **W3 & W4**
>
> W3:  hyperparameters sensitivity.
>
> We sincerely thank the reviewer for raising this question. In response, we would like to emphasize two key points:
>
> First, the hyperparameter values used in our experiments were fixed per model and were not individually tuned for each specific dataset. **As detailed in Appendix B.4, we provide the exact hyperparameter configurations for each model, ensuring repoducibility and transparency**. This practice demonstrates that our method does not require extensive, dataset-specific fine-tuning to be effective.
>
> Second, and more importantly, our comprehensive ablation studies, including **Figure.7, 8, 15 and 17**, were designed specifically to investigate the sensitivity and robustness of these choices. **In revised figures, we have clearly marked the performance baseline of the standard sampling method with a dotted line. The results consistently show that across a wide range of values for our hyperparameters, RF-Sampling consistently outperforms the standard baseline.**
>
> Thereofore, while an optimal setting exists, the method is not critically sensitive to small deviations from it. The primary conclusion from our ablation is not the existence of a single peak, but the existence of a broad effective region where RF-Sampling provides a reliable and significant improvement over the standard method. This demonstrates the practical robustness of our method.
>
> -----------------------------------------------------------------
>
> W4: writting confusion
>
> Thanks for pointing out this inconsistency. This has been corrected in the revised version. We apologize for this oversight and thank for the reviewer again for helping us improve the clarity and technical accuracy of our work.

---

> ### Author Response · Authors · 2025-11-21
> **Q1 & Q2**
>
> Q1: include a report on the inference time
>
> We sincerely thank the reviewer for raising this question. In our revised version, we have conducted experiments compared with other baseline methods under equivalent computation budget. As shown in Table.11, the inference time for RF-Sampling is highly comparable to all other baseline methods that also use 84 NFEs. Under this fair budget, RF-Sampling not only maintains comparable inference time but also achieves the top performance across almost all metrics, clearly showcasing its effectiveness.
>
> | Method | PickScore(↑) | HPSv2(↑) | AES(↑) | ImageReward(↑) | s/img(↓) |
> | :--- | ---: | ---: | ---: | ---: | ---: |
> | Standard (28 steps) | 21.99 | 29.32 | 5.9435 | 85.13 | 29.93 |
> | GI (28 steps) | 21.19 | 24.63 | 5.9534 | 28.94 | 31.33 |
> | CFG++ (28 steps) | 21.79 | 28.50 | 5.8821 | 85.17 | 32.46 |
> | CFG-Zero* (28 steps) | 21.88 | 29.37 | 5.9536 | 86.78 | 28.91 |
> | **Standard (28 × 3 = 84 steps)** | 21.96 | 29.60 | 5.9109 | 89.87 | 67.06 |
> | **GI (28 × 3 = 84 steps)** | 21.25 | 25.27 | 5.9335 | 28.16 | 67.04 |
> | **Z-Sampling (28 × 3 = 84 steps)** | 21.73 | 28.84 | 5.9091 | 89.03 | **65.00** |
> | **CFG++ (28 × 3 = 84 steps)** | 20.98 | 27.02 | 5.6144 | 64.73 | 68.07 |
> | **CFG-Zero\* (28 × 3 = 84 steps)** | **22.01** | 29.48 | 5.8949 | 97.22 | 65.47 |
> | **RF-Sampling** | 21.99 | **29.90** | **5.9981** | **101.50** | 65.04 |
>
> -------------------------------
> Q2: apply this sampling method by adjusting the CFG scale through that input
>
> We sincerely thank the reviewer for this insightful question regarding the potential of using the model's internal CFG scale input to achieve a similar effect. We have thoroughly investigated this possibility, and our empirical results confim that simply adjusting the pre-defined CFG scale $w$ is not a viable alternative to our RF-Sampling.
>
> As shown in Fig.16, we systematically varied the guidance scale $w$. The results demonstrate that deviating from the standard, recommended CFG scale, either by increasing or decreasing it, consistently leads to a degradation in image quality. This behavior is inherent to the design of CFG-distilled models. The CFG scale $w$ is a conditioning input that the model is trained to adhere to strictly. In constrast, the performance gains of RF-Sampling, as evidenced across all our main experiments, are therefore fundamentally distinct from and superior to the results obtained by naively adjusting the CFG scale.

---

> > ### Comment · Reviewer_7mEp · 2025-11-27
> >
> > Thank you for including the additional experiments.
> >
> > However, I still have concerns that the work is primarily an incremental extension of Z-Sampling with several empirical tricks, rather than being grounded in stronger theoretical basis. In addition, the empirical choice of many hyperparameters, and the slow speed further limit the practicality of the method.
> >
> > Considering these points, I will adjust my score to 4.

---

> > > ### Author Response · Authors · 2025-11-27
> > > **Response to the Follow-up Comment**
> > >
> > > We sincerely thank you for your continued engagement with our paper and for acknowledging the additional experiments included in our revision. We value your feedback and would like to take this opportunity to address your remaining concerns regarding the theoretical grounding, hyperparameter robustness, and inference speed.
> > >
> > > **1. Regarding the Theoretical Basis**
> > > We respectfully wish to clarify that our work is not merely an empirical extension of Z-Sampling. In our revision, we have provided a rigorous theoretical derivation of **RF-Sampling** (please see Appendix Sec.E). This theoretical analysis provides the mathematical justification for our approach, proving that it is grounded in a solid framework rather than being a collection of heuristic tricks. Our experimental results (Fig.2 and Fig.43) align closely with this theoretical analysis, serving as empirical validation of our derivations.
> > >
> > > **2. Regarding Hyperparameter Sensitivity and Robustness**
> > > We understand your concern regarding hyperparameters; however, our extensive ablation studies demonstrate that the method is highly robust and not dependent on precise tuning.
> > > *   As shown in **Figures 7, 8, 15, and 17**, we have clearly marked the performance baseline of the standard sampling method with a dotted line. ***The results demonstrate that RF-Sampling consistently outperforms the standard baseline across a **wide range** of hyperparameter values***.
> > > *   The primary conclusion from these figures is not the existence of a single, narrow optimal peak, but rather a "broad effective region." ***This indicates that our method is effective within a generous interval of choices and does not require brittle or exhaustive hyperparameter selection to work well in practice***.
> > >
> > > **3. Regarding Inference Speed**
> > > We appreciate you raising the issue of practicality, and we have addressed this in two ways:
> > > *   **Comparison with Z-Sampling:** In terms of computational cost, our method requires almost the same time as Z-Sampling yet achieves superior performance, making it a strictly better alternative in this context.
> > > *   **Compatibility with Acceleration:** To further address speed limitations, we demonstrated in **Table 3** that RF-Sampling is fully compatible with existing acceleration frameworks (Nunchaku). The results show that even when integrated with these acceleration methods to boost speed, our approach continues to outperform standard methods. This effectively mitigates the latency concern for practical applications.
> > >
> > > We hope these clarifications demonstrate that RF-Sampling is theoretically sound, robust to hyperparameter variations, and practically viable. We earnestly request that you reconsider your score based on these clarifications. Thanks for your efforts again.
> > >
> > > ***

---

### Official Review · Reviewer_Wj7i · 2025-11-04

**Soundness:** 3
**Presentation:** 4
**Contribution:** 3
**Rating:** 8
**Confidence:** 5

**Summary:**

The paper introduces RF-Sampling, a new inference-time sampling technique that enables trading off computational cost and output quality. The authors take inspiration from Z-Sampling, which introduces a process where a denoising step with a high guidance scale is followed by an inversion step with a low guidance scale to better align the noise with the desired semantics before proceeding with standard denoising. They adapt this idea for CFG-distilled flow matching models and propose a similar approach that applies CFG-like guidance directly on the text embeddings instead. The method is comprehensively evaluated across several text-to-image, text-to-video, and image-editing models, showing improved performance compared to prior techniques.

**Strengths:**

- The paper is very well written and easy to follow.
- The proposed algorithm in simple to understand and easy to implement.
- The experiment suite is quite extensive with an impressive number of models, benchmarks, and ablations.

**Weaknesses:**

- Since the proposed algorithm can be viewed as an inference-time scaling method, including a simple Best-of-$N$ baseline in the experiments would help better position the paper within the literature. For instance, according to Figure 17, performing RF-Sampling on all steps appears to yield the best results. Using the $\alpha = 1$ setting used in the experiments, RF-Sampling requires three times as many forward passes as generating a single sample from the base model. Therefore, comparing this approach against a Best-of-3 baseline from the base model would be a useful addition.
- The paper argues that Z-Sampling cannot be directly applied to CFG-distilled models such as FLUX. However, it is unclear why this would be the case, since these models typically distill a range of guidance values during training. As a result, one could, in principle, perform a variant of Z-Sampling by simply changing the guidance scale values during the denoising and inversion steps.
- Following the previous point, Section 3.2 (Lines 203–205) states: “Flow models are typically trained only under conditional settings (Labs, 2024; Daniel Verdú, 2024). As a result, directly using CFG techniques or adopting an empty-text embedding as guidance for flow models is inappropriate.” According to the cited literature [1], this claim is inaccurate. To obtain a guidance-distilled model such as FLUX-dev, a standard flow matching model is first trained with text embedding dropout to enable guidance. This model is then CFG-distilled into a student that takes the guidance scale as input and outputs the guided velocity in a single forward pass.

[1] Meng, Chenlin, et al. "On distillation of guided diffusion models." Proceedings of the IEEE/CVF conference on computer vision and pattern recognition. 2023.

**Questions:**

1. Could the authors clarify why Z-sampling cannot be applied for FLUX-dev?

---

> ### Author Response · Authors · 2025-11-21
> **Weakness 1: Comparison with Best-of-N method.**
>
> We sincerely thank the reviewer for this insightful suggestion. We agree that comparing with Best-of-N sampling is crucial for positional our method within the inference-time scaling literature. Following your advice, we have conducted comprehensive comparisons with Best-of-N baselines, as well as other methods under an equivalent computational buget.
> These new results are now included in the revised manuscript as Table.10 and Table.11.
>
> ● As shown in Table.10, we directly compare our RF-Sampling with the Best-of-N method, where we use the HPSv2 as the verifier model. The results demonstrate that RF-Sampling achieves a superior trade-off between performance and efficiency. Specifically, while Best-of-5 achieves the highest scores, it requires more than double the inference time per image compared to RF-Sampling (about six time per image compared to standard inference). More importantly, RF-Sampling outperforms Best-of-3 across almost metrics while being approximately 1.5 times faster. This validate that our method is more efficient than simply generating multiple independent samples and selecting the best one.
>
> | Method | PickScore(↑) | HPSv2(↑) | AES(↑) | ImageReward(↑) | s/img(↓) |
> | :--- | ---: | ---: | ---: | ---: | ---: |
> | Standard (28 steps) | 21.99 | 29.32 | 5.9435 | 85.13 | 29.93 |
> | Standard (28 × 3 = 84 steps) | 21.96 | 29.60 | 5.9109 | 89.87 | 67.06 |
> | Best-of-5 | 22.21 | 30.58 | 5.9849 | 106.69 | 154.17 |
> | Best-of-3 | 21.94 | **30.14** | 5.9642 | 100.40 | 97.63 |
> | RF-Sampling | **21.99** | 29.90 | **5.9981** | **101.50** | 65.04 |
>
> ● Furthermore, Table. 11 provides a detailed comparison under an equivalent computational budget (equivalent to 28 x 3 = 84 NFEs). The results how that RF-Sampling consistently outperforms almost all baseline methods across different evaluation metrics while maintaining a comparable inference time. This firmly establishes the effectiveness of our method in a fair computational setting.
>
> | Method | PickScore(↑) | HPSv2(↑) | AES(↑) | ImageReward(↑) | s/img(↓) |
> | :--- | ---: | ---: | ---: | ---: | ---: |
> | Standard (28 steps) | 21.99 | 29.32 | 5.9435 | 85.13 | 29.93 |
> | GI (28 steps) | 21.19 | 24.63 | 5.9534 | 28.94 | 31.33 |
> | CFG++ (28 steps) | 21.79 | 28.50 | 5.8821 | 85.17 | 32.46 |
> | CFG-Zero* (28 steps) | 21.88 | 29.37 | 5.9536 | 86.78 | 28.91 |
> | **Standard (28 × 3 = 84 steps)** | 21.96 | 29.60 | 5.9109 | 89.87 | 67.06 |
> | **GI (28 × 3 = 84 steps)** | 21.25 | 25.27 | 5.9335 | 28.16 | 67.04 |
> | **Z-Sampling (28 × 3 = 84 steps)** | 21.73 | 28.84 | 5.9091 | 89.03 | **65.00** |
> | **CFG++ (28 × 3 = 84 steps)** | 20.98 | 27.02 | 5.6144 | 64.73 | 68.07 |
> | **CFG-Zero\* (28 × 3 = 84 steps)** | **22.01** | 29.48 | 5.8949 | 97.22 | 65.47 |
> | **RF-Sampling** | 21.99 | **29.90** | **5.9981** | **101.50** | 65.04 |
>
> We believe these new experiments and analyses have significantly strengthened our paper by directly addressing your concern and clearly demonstrating the advantages of RF-Sampling.

---

> ### Author Response · Authors · 2025-11-21
> **W2 & Q1: Could the authors clarify why Z-sampling cannot be applied for FLUX-dev**
>
> We sincerely thank the reviewer for this insightful question, which allows us to clarify a crucial technical distinction between standard diffusion models and CFG-distilled models like FLUX. The reviewer is correct that CFG-distilled models are trained with a range of guidance scales, but the core issue lies in the fundamental difference in how they use this scale compared to a standard model that natively supports CFG [1].
>
> The key point is that a CFG-distilled model does not have a true unconditional generation pathway. In a standard diffusion model, CFG works by explicitly combining a conditional forward and an unconditional forward pass. In constrast, a CFG-distilled model is trained to mimic the output of this two-pass CFG process in a single forward pass [2], where the guidance scale $w$ is provided as an input condition. Therefore, for a CFG-distilled model, even setting $w=1$ does not yield an unconditional output. It simply generates an image conditioned on $w=1$. Its output is always a function of the provided guidance scale and cannot be decoupled from it.
>
> Our empirical evidence strongly supports this. As shown in Fig.42, when we set $w=1$ for FLUX-Lite, the generated images are clearly semantically aligned with the input text prompt, comfirming that the output remains conditionally generated.
>
> This fundamental architectural difference makes it theoretically invalid to directly apply Z-Sampling, W2SD-Sampling or similar methods to CFG-distilled models. These methods rely on the clear separation between conditional and unconditional paths in standard CFG to perform their inversion operations. Manipulating the $w$ input of a CFG-distilled model does not replicate this behavior because it does not correspond to traversing between conditional and unconditional states, but rather between different conditioned states.
>
> As suggested by the reviewer, we tried to adapt Z-Sampling and W2SD-Sampling for CFG-distilled models like FLUX-Lite and FLUX-Dev, the results are already presented in Appendix Fig. 12., and we also provde the discussion in Appendix. C. The results demonstrate that for those CFG-based sampling methods like Z-Sampling and W2SD-Sampling, due to the lack of unconditional model forward pass on CFG-distilled models, can not work.
>
> [1]Classifier-free diffusion guidance
>
> [2]On Distillation of Guided Diffusion Models

---

> ### Author Response · Authors · 2025-11-21
> **W3：**
>
> We sincerely thank the reviewer for pointing out this.  We have corrected this writing issue in the revised version.

---

### Author Response · Authors · 2025-11-21
**General Response**

We thank all the reviewers and the Area Chair for their insightful comments and constructive suggestions. We are very glad that the reviewers find our work "timely and meaningful". Based on the valuable feedback, we have revised our manuscript and conducted extensive new experiments. The key revisions are summarized as follows:

1. Refined the explanation of why Z-Sampling, W2SD-Sampling and other CFG-based methods cannot be applied to CFG-distilled models like FLUX.

2. Added comparisons with new baseline methods, including Best-of-N and Ma et al. (CVPR 2025), with detailed inference time consumption reports.

3. Added breakdown details for Fig. 2, providing specific time consumption and step counts, along with comprehensive baseline comparisons under equivalent computational budgets.

4. Added FID and IS evaluations on ImageNet-1K.

We hope that our replies and revisions have satisfactorily addressed the reviewers' questions and further strengthened their assessment of our work. All revised content in the manuscript has been highlighted in blue for easy identification. We remain open to further discussion should any issues remain.

---

### Author Response · Authors · 2025-11-25
**Theoretical Discussion of RF-Sampling as Gradient Ascent**

In our latest revised version, we conduct the theoretical discussion of RF-Sampling. In **Appendix Sec.E**, we rigorously prove that RF-Sampling is an approximate **gradient ascent on the alignment score** $J(x) = \log p(c|x)$.

1.  **Gradient Proxy Construction:**
    We derive that the difference between conditional and unconditional velocity fields is proportional to the alignment gradient $\nabla_x J(x)$. Through Taylor expansion of the embeddings, we prove that our "reflection" operation ($\Delta_{RF}$) cancels out baseline flow components, yielding a displacement strictly proportional to this gradient:
    $$ \Delta_{RF} \approx \delta t \cdot \mathcal{A} \cdot \nabla_x J(x) $$

2.  **First-Order Analysis (Validity & Time-Scaling):**
    Since $\Delta_{RF}$ aligns with $\nabla_x J(x)$, the update $x + \gamma \Delta_{RF}$ guarantees a positive increase in the objective. This theoretically explains the **monotonic quality improvement** observed with increased inference steps (Fig. 2): more steps imply a smaller $\delta t$, which reduces the approximation error of the Taylor expansion ($O(\delta t^2)$), resulting in a more precise gradient estimate.

3.  **Second-Order Analysis (Curvature & Merge Ratio):**
    By expanding the objective to the Hessian term (Second-Order), we define the improvement as a parabolic function:
    $$ \Delta J(\gamma) \approx \gamma \cdot (\text{Linear Gain}) - \frac{1}{2}\gamma^2 \cdot (\text{Curvature Penalty}) $$

    This mathematically predicts the **inverted U-shaped performance curve** for the merge ratio $\gamma$ (Fig. 43). The optimal $\gamma^*$ exists as a trade-off: it must be large enough to maximize the linear gain but bounded by the local curvature of the semantic manifold to prevent degradation.

We hope that our replies and revisions have satisfactorily addressed the reviewers' questions and further strengthened their assessment of our work. All revised content in the manuscript has been highlighted in blue for easy identification. We remain open to further discussion should any issues remain.

---

### Author Response · Authors · 2025-11-28

Dear Reviewer:

We hope this message finds you well. As the discussion period is nearing its end, we hope that we have addressed all your concerns satisfactorily. If there are any additional points or feedback you'd like us to consider, please let us know. Your insights are invaluable to us, and we're eager to address any remaining issues to improve our work.

Thank you for your time and effort in reviewing our paper.

---

### Meta-Review · Area_Chair_SNbw · 2026-01-11

**Summary:**

Reviewers appreciated the simplicity of the proposed algorithm and experimental results, but also raised several issues with the manuscript including, comparison with Best-of-3 baseline from the base model, missing theoretical justification, missing hyperparameter sensitivity analysis, concerns with respect to the proposed approach being an incremental extension of Z-Sampling with several empirical tricks, and missing mathematical foundations of convergence or guarantees.

**Reviewer Concerns:**

The meta-reviewer believes that the rebuttal addresses some of the concerns of the reviews such as, comparison with Best-of-N method and hyperparameters sensitivity). However, the meta-reviewer believes several of the concerns, initial raised by the reviewers, remained insufficiently addressed including, a detailed theoretical basis of RF-Sampling and inference speed comparison. This has been voiced by one of the reviewers even post-rebuttal. Consequently, three reviewers were largely negative about the manuscript.

**Reviewer Scores:**

The meta-reviewer believes the rebuttal only partially addressed the reviewer's concerns. For instance, the issue of detailed theoretical basis of RF-Sampling remained (even voiced by one of the reviewers post-rebuttal). Majority (3) of the reviewers generally remained negative highlighting multiple and somewhat overlapping issues. Based on the reviewer's comments and rebuttal, the meta-reviewer believes the negative concerns are comprehensive and are not fully resolved in the rebuttal.

---

### Decision · Program_Chairs · 2026-01-26

Reject